# Promising Results of Associating Liver Partition and Portal Vein Ligation for Staged Hepatectomy for Perihilar Cholangiocarcinoma in a Systematic Review and Single-Arm Meta-Analysis

**DOI:** 10.3390/cancers16040771

**Published:** 2024-02-13

**Authors:** Mohammad Golriz, Ali Ramouz, Ahmed Hammad, Ehsan Aminizadeh, Nastaran Sabetkish, Elias Khajeh, Omid Ghamarnejad, Carlos Carvalho, Hugo Rio-Tinto, De-Hua Chang, Ana Alagoa Joao, Gil Goncalves, Arianeb Mehrabi

**Affiliations:** 1Department of General, Visceral and Transplantation Surgery, University of Heidelberg, 69120 Heidelberg, Germany; mohammad.golriz@med.uni-heidelberg.de (M.G.); ali.ramouz@med.uni-heidelberg.de (A.R.); ahmed.hammad@med.uni-heidelberg.de (A.H.); ehsan.aminizadeh@med.uni-heidelberg.de (E.A.); nastaran.sabetkish@med.uni-heidelberg.de (N.S.); elias.khajeh@med.uni-heidelberg.de (E.K.); omid.ghamarnejad@med.uni-heidelberg.de (O.G.); 2Liver Cancer Centre Heidelberg (LCCH), University Hospital Heidelberg, 69120 Heidelberg, Germany; de-hua.chang@med.uni-heidelberg.de; 3Clinic of General and Visceral Surgery, Diakonie in Südwestfallen, 57076 Siegen, Germany; 4Digestive Oncology Unit, Champalimaud Foundation, 1400-038 Lisbon, Portugal; carlos.carvalho@fundacaochampalimaud.pt; 5Department of Radiology, Champalimaud Foundation, 1400-038 Lisbon, Portugal; hugo.rio-tinto@fundacaochampalimaud.pt; 6Department of Diagnostic and Interventional Radiology, University Hospital Heidelberg, 69120 Heidelberg, Germany; 7Hepato-Pancreato-Biliary Surgery Unit, Department of Digestive Surgery, Champalimaud Foundation, 1400-038 Lisbon, Portugal; ana.joao@fundacaochampalimaud.pt (A.A.J.); gil.goncalves@fundacaochampalimaud.pt (G.G.)

**Keywords:** hepatectomy, ALPPS, perihilar cholangiocarcinoma, morbidity, mortality

## Abstract

**Simple Summary:**

The increasing popularity of ALPPS (Associating Liver Partition and Portal vein Ligation for Staged hepatectomy) in treating unresectable liver tumors has extended to perihilar cholangiocarcinoma (phCC), despite some reservations about its use in these cases. This systematic review and pooled data analysis, which included 112 phCC patients from 18 studies, was based on the literature from MEDLINE and Web of Science up to December 2023. The findings showed significant rates of major morbidity (43%) and mortality (22%), with a post-hepatectomy liver failure (PHLF) rate of 23%. However, promising one-year disease-free and overall survival rates were observed at 65% and 69%, respectively. This study concludes that ALPPS offers a viable treatment option for phCC, potentially superior to alternatives, although it is associated with considerable risks. Improvements in surgical techniques and perioperative management could further enhance its efficacy and safety.

**Abstract:**

Background: ALPPS popularity is increasing among surgeons worldwide and its indications are expanding to cure patients with primarily unresectable liver tumors. Few reports recommended limitations or even contraindications of ALPPS in perihilar cholangiocarcinoma (phCC). Here, we discuss the results of ALPPS in patients with phCC in a systematic review as well as a pooled data analysis. Methods: MEDLINE and Web of Science databases were systematically searched for relevant literature up to December 2023. All studies reporting ALPPS in the management of phCC were included. A single-arm meta-analysis of proportions was carried out to estimate the overall rate of outcomes. Results: After obtaining 207 articles from the primary search, data of 18 studies containing 112 phCC patients were included in our systematic review. Rates of major morbidity and mortality were calculated to be 43% and 22%, respectively. The meta-analysis revealed a PHLF rate of 23%. One-year disease-free survival was 65% and one-year overall survival was 69%. Conclusions: ALPPS provides a good chance of cure for patients with phCC in comparison to alternative treatment options, but at the expense of debatable morbidity and mortality. With refinement of the surgical technique and better perioperative patient management, the results of ALPPS in patients with phCC were improved.

## 1. Introduction

Cholangiocarcinomas account for approximately 3% of all gastrointestinal malignancies and are the second-most common primary hepatic malignancy, with a five-year survival rate of up to 20% depending on the therapeutic modality [1,2,3]. Over the past two decades, the incidence of cholangiocarcinoma has increased worldwide, which can be attributed to the accompanying increase in predisposing factors such as alcoholic liver disease, hepatitis C virus infection, and cirrhosis [4,5]. Cholangiocarcinomas are divided into three main types: intrahepatic, perihilar, and distal. Perihilar cholangiocarcinoma (phCC) constitutes about 60% to 70% of cholangiocarcinomas followed by the distal and then the intrahepatic types [6]. Liver resection is the only potential curative option for patients with cholangiocarcinoma, which increases the five-year survival rate up to 25–30% in all cases and to 30–60% in phCC cases, if amenable to surgery [2,6].

However, only 10–40% of tumors can be resected at the time of diagnosis [6]. The majority of patients are considered unresectable due to insufficient future remnant volume (RLV) with a high possibility of developing post-hepatectomy liver failure (PHLF) [7,8]. To increase the RLV, different modalities such as portal vein embolization (PVE), two-stage liver resection, and associating liver partition and portal vein ligation for staged hepatectomy (ALPPS) have been introduced [7]. Although several studies showed increased volume and improved surgical outcomes after ALPPS for other indications, using ALPPS for phCC is still a matter of controversy. Most of the studies, including the recent systematic review of ALPPS in cholangiocarcinoma, focus only on intrahepatic and not perihilar cholangiocarcinoma [9]. Some surgeons believe that ALPPS may not be performed for phCC patients. This is based on a few studies, which showed inferior outcomes of ALPPS for phCC compared to standard resection or other indications, such as colorectal liver metastases and hepatocellular carcinoma [10,11,12]. 

On the other hand, some recent studies have revealed encouraging short- and long-term outcomes for ALPPS utilization in patients with primary unresectable phCC, comparable to other hepatic malignancies. In order to address the ongoing debate and evaluate the current study, the present systematic review aimed to evaluate and discuss the surgical and oncological outcomes of all reported ALPPS procedures for phCC. 

## 2. Materials and Methods

This study was conducted according to recommendations of the Study Center of the German Society of Surgery and Preferred Reporting Items for Systematic Reviews and Meta-Analyses (PRISMA) guidelines [13,14]. The protocol has not been registered.

### 2.1. Eligibility Criteria

The research question was formulated based on the Population, Intervention, Comparison, Outcome, and Study design (PICOS) strategy. Accordingly, the inclusion criteria were as follows:

*Population:* all patients with phCC undergoing ALPPS procedure.

*Intervention:* all types of ALPPS procedure.

Comparator: none.

*Outcome:* dropout, postoperative morbidity, mortality, and recurrence.

*Study design:* any study design except case reports, narrative or systematic reviews, study protocols, experimental or animal studies, conference abstracts, letters, and common overviews.

To prevent repeat analysis of the same patients, articles were carefully reviewed and double publications from the same center and overlapping reports were excluded.

### 2.2. Literature Search

Medline (via PubMed), Web of Science, and CENTRAL databases were systematically searched to identify relevant articles published up to December 2023. The search was not limited to the study type, language, and publication date. The following keyword combinations were used: (“Associating liver partition and portal vein ligation for staged hepatectomy” OR “ALPPS” OR “In-situ liver splitting” OR “In-situ parenchymal division” OR “Associating liver tourniquet and portal vein ligation for staged hepatectomy” OR “ALTPS” OR “Radio-frequency assisted liver partition and portal vein ligation” OR “RALPP” OR “p-ALPPS” OR “laparoscopic microwave ablation and portal vein ligation for staged hepatectomy” OR “LAPS” OR “mini-ALPPS” OR “combined in-situ splitting of the left lateral liver lobe with postoperative right portal vein embolization”) AND (“cholangiocarcinoma*” OR “cholangiocellular carcinoma*” OR “bile duct cancer*” OR “klatskin” OR “bile duct carcinoma*” OR “bile duct neoplasm*” OR “biliary carcinoma*” OR “biliary neoplasm*” OR “biliary cancer*” OR “biliary tree cancer*” OR “biliary tree carcinoma*” OR “biliary tract cancer*” OR “biliary tract carcinoma*” OR “biliary tree neoplasm*” OR “biliary tract neoplasm*”). Reference lists from all eligible articles were also screened for studies that were not identified by the literature search.

### 2.3. Study Selection and Data Extraction

Two authors (EK and AH) independently screened and selected all titles and abstracts based on the predefined PICOS eligibility criteria. Eligible full-text articles were reviewed by OG and EA, who independently appraised the articles and extracted their data. Discrepancies between the two authors were resolved by discussion with senior authors (MG and AM). The study type, sample size, demographic data (country, age, and gender), type of ALPPS, type of resection, additional surgical procedures, and short- (PHLF, morbidity, and mortality) and long-term outcomes (e.g., 1 year overall and disease-free survivals) were extracted.

### 2.4. Quality Assessment (Bias)

The quality of each study was assessed by two independent authors (EK and OG) using the methodological index for non-randomized studies (MINORS) established by Slim et al. [15] Disagreements were resolved by discussion and consensus. Quality was determined based on eight MINORS items. The items were scored as 0 (not reported), 1 (reported but inadequate), or 2 (reported and adequate). Studies with 12 or more points were considered high-quality. Studies with 8–12 points were considered intermediate-quality. Studies with less than 8 points were considered low-quality. The Grading of Recommendations Assessment, Development and Evaluation (GRADE) approach was also used to evaluate the overall quality of the evidence of included studies [16].

### 2.5. Definition of Extracted Data

#### 2.5.1. Demographic and Baseline Characteristics

General study information, including year and period of the study, number of cases, country, and study design, were collected, as well as the mean age of the patients and underlying liver pathology, accounting for ALPPS procedure.

#### 2.5.2. Surgical Technique

For surgical technique, we recorded whether the ALPPS procedure was classic or modified, as well as the type of modification. We also recorded additional details of the surgical intervention, such as liver resection type and performing a hepaticojejunostomy at stage I or II.

#### 2.5.3. Postoperative Complications and Mortality

Morbidity was defined as any complication occurring after stage 1 and stage 2. Furthermore, the rate of major morbidities after stage I and stage II was defined as ≥grade III based on the Clavien–Dindo classification [17]. PHLFs were not even defined or classified based on the ISGLS [18] and 50–50 criteria [19]. The mortality was defined in 10 studies as up to 90 days/in-hospital mortality after ALPPS procedure.

#### 2.5.4. Oncological Outcomes

Oncological outcomes included one-year disease-free survival (DFS) and one-year overall survival (OS). Disease-free survival was considered as the patient survival without any signs or symptoms of the primary cancer for one year after ALPPS procedure. One-year OS was defined as patient survival, regardless of the underlying disease status, in the one-year period after ALPPS procedure.

### 2.6. Statistical Analysis

For single-arm meta-analysis, proportions were calculated using a random-effects model to generate pooled rates and their confidence intervals (CIs) using per-protocol and intention-to-treat data when available. Summary effect measures are presented along with their corresponding 95% CIs. Statistical heterogeneity was assessed using χ2 and inconsistency analyses, with the threshold for heterogeneity considered present if the *p*-value was lower than 0.05 or the I^2^ was greater than 50%. Publication bias was assessed by means of a funnel plot. The R software version 4.3.2^©^ (R Foundation for Statistical Computing, Vienna, Austria) and Meta package were used for data analysis. In the provided forest plots, each horizontal line is accompanied by a central square, which signifies the point estimate of the effect in a specific study’s CI. The box’s size is indicative of the study’s weight concerning the pooled estimate. The overall effect estimate of the meta-analysis is symbolized by a rhomb. The rhomb’s center on the x-axis indicates the point estimate, while its width illustrates the 95% CI around the pooled effect’s point estimate.

## 3. Results

### 3.1. Characteristics of Studies

The literature search yielded 207 articles, which were included for primary screening. After removal of 63 duplicates, 144 articles were reviewed. Of these, 105 articles were excluded for various reasons, such as irrelevant studies, experimental studies, review articles, or letters to the editor. After reading the remaining 38 articles, 20 articles were excluded, because 6 of them had missing specific outcome data, 5 studies were duplicate reports from the same institution/center, and 5 studies reported only the surgical technique. Furthermore, four articles were excluded as the authors analyzed data from the international ALPPS registry (Figure 1). 

In addition, the eight patients reported by Mehrabi et al. [20] were also excluded, since they were analyzed by Balci et al. [21]. Finally, eighteen articles were included in the qualitative and quantitative analyses [7,10,20,21,22,23,24,25,26,27,28,29,30,31,32,33,34,35] (Figure 1). The characteristics of the included studies are presented in Table 1.

### 3.2. Quality Assessment

Detailed assessments of the included studies are shown in Table 2. Eleven of the included articles presented intermediate-quality evidence, while the remaining studies presented low-quality evidence. The six studies were of low quality because they were retrospective, had a low sample size, and lacked a comparison with alternative techniques like PVE or conventional two-stage hepatectomy. Also, it was not feasible to conduct fully blinded studies for this research question, since both patients and surgeons knew the nature of the ALPPS technique. Therefore, the studies were considered as unblinded. Furthermore, according to the GRADE approach, the overall quality of evidence was low among the included studies.

### 3.3. Demographic and Baseline Characteristics

Overall, 18 studies with 372 patients, aged between 35 and 77 years old that underwent ALPPS for all indications, were included. Among these patients, colorectal metastases turned out to be the most common indication for the ALPPS procedure (146 patients (39.1%)). phCC (112 patients (30.1%)) and other metastases (42 patients (11.4%)) were the second- and third-leading indications for the ALPPS procedure. In total, 112 patients with phCC underwent the ALPPS procedure (Table 1). The indication for ALPPS in each study is summarized in Appendix A.

### 3.4. Surgical Technique

In eleven studies, the classic ALPPS technique was performed, while partial ALPPS was performed in three studies, and partial transileocecal portal vein embolization ALPPS (TIPE ALPPS), percutaneous radiofrequency-assisted ALPPS (PRALPPS), softALPPS, and partial laparoscopic first-stage ALPPS were each performed in one study. Five studies did not report the side and amount of resection. Only in one study did two patients (1.8%) not proceed to stage II, due to a diagnosis of peritoneal carcinomatosis [31]. Fifteen studies reported on hepaticojejunostomy. Of eleven studies reporting the time of hepaticojejunostomy, the anastomosis was placed during stage I in five studies (45.4%) and during stage II in five studies (45.4%) (Table 3). Only in one multicentric study (9.2%) was the hepaticojejunostomy carried out in both stages. R0 status following ALPPS was achieved in 83.7% (87 of 104) of patients, with the majority of studies (60%) reporting 100% for the R0 resection margin (Table 4).

### 3.5. Postoperative Complications and Mortality

Ten and eleven studies reported the post-ALPPS morbidities after stage I and II, respectively. The major morbidity rate for patients with phCC following Stage I of the ALPPS procedure ranged from 0 to 50% in different studies. The meta-analysis showed an estimated rate of 24% (95% CI = 16–35%; I^2^-heterogeneity = 0%, *p*-value = 0.92) for major morbidity after ALPPS stage I. Subsequently, the pooled rate of major morbidity after stage II was 43% (95% CI = 30–58%; I^2^-heterogeneity = 0%, *p*-value = 0.59), ranging between 0 and 100% (Figure 2). The most common complications were biliary leakage, abdominal sepsis, and peritonitis. PHLF was reported in 11 studies (Figure 3).

Data regarding the definition and grade of PHLF in different studies have been provided in Appendix A. The PHLF rate (grade B and C) was between 0 and 33%. In total, the estimated rate of PHLF after the ALPPS procedure was calculated to be 23% (95% CI = 15–34%; I^2^-heterogeneity = 0%, *p*-value = 0.99).

Postoperative mortality was reported in 16 studies. The mortality rate following ALPPS in phCC patients varied between 0 and 60%. Among 104 patients, the mortality rate was defined as follows: in-hospital mortality in one study [36], early postoperative mortality (range from 9 to 36 days) in one study [22], and 90-day mortality after the operation in three studies [23,27]. Ten studies reported 0% mortality [7,10,26,28,29,30,31,33,34,35]. The meta-analysis estimated a mortality rate of 22% (95% CI = 14–34%; I^2^-heterogeneity = 0%, *p*-value= 0.64) (Figure 4). The cause of mortality has been provided in detail in Appendix A.

### 3.6. Oncological Outcomes

One-year DFS (DFS) was reported in eight studies, with a pooled rate of 65% (95% CI = 47–79%; I^2^-heterogeneity = 14%, *p*-value = 0.32) using a single-arm meta-analysis (Figure 5). Finally, the one-year OS was reported in nine studies (Figure 6). The meta-analysis estimated the rate of one-year OS after the ALPPS procedure to be 69% (95% CI = 58–79%; I^2^-heterogeneity = 0%, *p*-value = 0.79).

## 4. Discussion

Surgical resection remains the only definitive therapy for many patients with primary or metastatic liver tumors. In spite of new developments in perioperative management as well as intraoperative methods, there are still many patients who cannot be operated on, due to the risk of PHLF. Many methods have been used to try to overcome this problem by giving the tumor-free liver a chance to regenerate while using the rest capacity of the tumor-involved liver [36]. Among these methods, ALPPS showed a better regeneration rate within a shorter time and accordingly with fewer drop-outs [7,37]. However, the few initial analyses of ALPPS results with regard to the indications showed inferior outcomes in cholangiocarcinoma patients compared to colorectal liver metastasis or hepatocellular carcinoma [12,38]. 

Although the outcomes of ALPPS in intrahepatic cholangiocarcinoma have been well discussed in a recent systematic review [9], the ALPPS in phCC as the most common type of cholangiocarcinoma has not been systematically discussed so far. The only available evaluation is the comparison between 29 ALPPS in phCC following only 4 years of experience and 29 standard resections following more than 16 years of experience. This study reported a mortality rate of 48% following ALPPS in phCC patients compared to 13% following standard liver resection [12]. Since then, many attempts have been performed to optimize the initial results, including the better and careful selection of patients, as well as technical modifications, such as radio-frequency-assisted ALPPS (RALPPS), tourniquet ALPPS, partial-ALPPS, laparoscopic ALPPS, mini-ALPPS, and hybrid ALPPS [39,40,41,42,43,44]. The main aim of all these procedures was to minimize the first step of the operation as much as possible. Also, it was suggested that this complex procedure has to be limited to highly specialized centers with highly trained surgeons [45]. Moreover, the release of the ALPPS risk score which aimed to assist the clinician in reducing ALPPS-related early mortality could help in this regard [46]. 

On this basis, we summarized the outcomes of ALPPS in phCC patients and performed a systematic review with a pooled data analysis. Primarily, we believe that the outcome of ALPPS in phCC may not be compared with the outcome of the patients who underwent standard liver resection, because, if this option were available to the patients and the remnant liver volume was large enough, an alternative method would not be discussed. Also, the alternative of these patients, which is palliative therapy, shows around 90% mortality in 5 years [2]. Moreover, it is obvious that the ALPPS procedure shows better outcomes in CRLM and HCC in comparison to phCC [11]. But it must be considered that, without any intervention, the survival of phCC patients is significantly lower and ALPPS is still one of the best therapeutic options. 

The initial reported outcomes of ALPPS came mostly from CRLM patients with better preoperative conditions. The report from the ALPPS registry in 2014 analyzed the results of 202 ALPPS operations, from which 106 were hemihepatectomies. The future liver remnant volume range was 252–421 cm^3^ before the first stage [38]. Even in the report from the ALPPS registry in 2018, there were 183 hemihepatectomies [47]. In comparison, approximately all reported cases of ALPPS in phCC underwent trisectionectomy. Also, the surgical experience with the ALPPS procedure was not that advanced at the time of initial reports. In the report of the ALPPS registry from 2014, there were only 11 perihilar CCs [38]. In the further report in 2016, there were 29 cases, and only 7 out of 23 participating centers had experience with more than 16 ALPPS procedures for all indications [12]. In addition, the role of preoperative cholestasis and cholangitis in deteriorating the patients’ outcomes was not taken into account, which was later addressed by the ALPPS risk score [46].

Our review of the published studies in the literature with 112 ALPPS procedures in phCC patients shows 43% and 22% morbidity and mortality rates overall. A deeper analysis of the data [21], considering better patient selection including preparation and management of the cholestasis and infection as well as reducing the first-stage operation, shows comparable results for ALPPS in phCC to other indications [21]. Technically, ALPPS is also a feasible method in patients with phCC near the biliary confluence, since hepaticojejunostomy after confluence resection was carried out almost in 70% of the cases. Particularly, among patients with phCC suffering from preoperative cholestase and cholangitis, a delayed bile duct reconstruction during the second stage of ALPPS is shown to diminish the rate of morbidity and mortality [20]. Meanwhile, technical modifications have been proposed to optimize the postoperative results. Balci et al. described laparoscopic partial ALPPS in stage one and reported 0% morbidity and mortality rates following his method [30]. Sakamoto et al. described transileocecal portal vein embolization combined with ALPPS (partial TIPE ALPPS) for perihilar cancer with 0% mortality [48]. In our analysis, major postoperative complications ranged from 0% to 100% in classic ALPPS patients [10,23,35], 50% in partial TIPE ALPPS [28], 50% in partial ALPPS [29], and 0% in laparoscopic first-stage partial ALPPS [30]. Regarding the above-mentioned results, it can be postulated that better patient management and reducing the extent of surgery in step one can lead to promising results. Also, from an oncological point of view, completing tumor resection with a higher rate of R0 resection plays a major role in long-term survival outcomes [49]. Our review showed that the rate of R0 resection in phCC patients with ALPPS was achieved in a range from 71 to 100%. These results are comparable to the results of standard extended hepatectomies [50,51].

The unfavorable outcomes of the ALPPS procedure in regard to the postoperative morbidity and mortality in CC patients affected mainly phCC patients [38]. This could be attributed to the complexity of surgical procedures in phCC with the possible necessity of vascular resection besides the need for hepaticojejunostomy. Moreover, the presence of longstanding biliary obstruction reduces the liver capacity to regenerate and increase the patient’s susceptibility to postoperative biliary infection following interventional drainage. These can lead to intraabdominal sepsis and worse postoperative outcomes [12,52]. Therefore, despite the remarkable hypertrophy of FLR after ALPPS procedure, the functional capacity of hepatocytes in FLR is still questioned besides the immaturity of canalicular ductule networks in the FLR, which, in addition to the presence of cholestasis after major resection, subsequently predisposes to PHLF and mortality [53,54]. These highlight the role of careful patient selection and surgical technique as well as preoperative management. 

In addition, recent studies have highlighted the advantageous role of minimally invasive surgery in treating patients with phCC. A recent series of laparoscopic and robot-assisted liver resections among these patients revealed no major complications and mortality [55,56]. Based on this evidence, minimally invasive techniques can be considered safe and feasible approaches, at least for the first stage of ALPPS, to reduce the extent of the procedure and minimize potential complications. Furthermore, numerous advanced modalities, such as augmented reality and enhanced tumor navigation via biophotonics (such as indocyanine green), have been integrated into liver surgery, yielding improved results [57,58]. Particularly within the realm of robotic surgery, these options can enhance their utilization and amplify their benefits [57]. Potential advantages encompass reduced morbidity and enhancements in oncological outcomes via improved tumor navigation [57,58]. However, recent surveys have revealed certain limitations, including inadequate technological development and a lack of clinical evidence [59].

The main limitation of this review was the quality of original data, since 11 of the included articles presented intermediate-quality evidence and the remaining six presented low-quality evidence. The studies were of low quality because they were either retrospective, had a low sample size, or lacked a comparison with alternative techniques.

## 5. Conclusions

In conclusion, ALPPS enables the surgical treatment of phCC patients with extremely low FLR volume and gives them a chance of cure. In spite of primarily questionable results, refinements in the whole process of this procedure including patient selection, technique modifications, as well as postoperative management are considered to result in better outcomes. Further investigations through prospectively established multicenter randomized studies are required for better confirmation of these results.

## Figures and Tables

**Figure 1 cancers-16-00771-f001:**
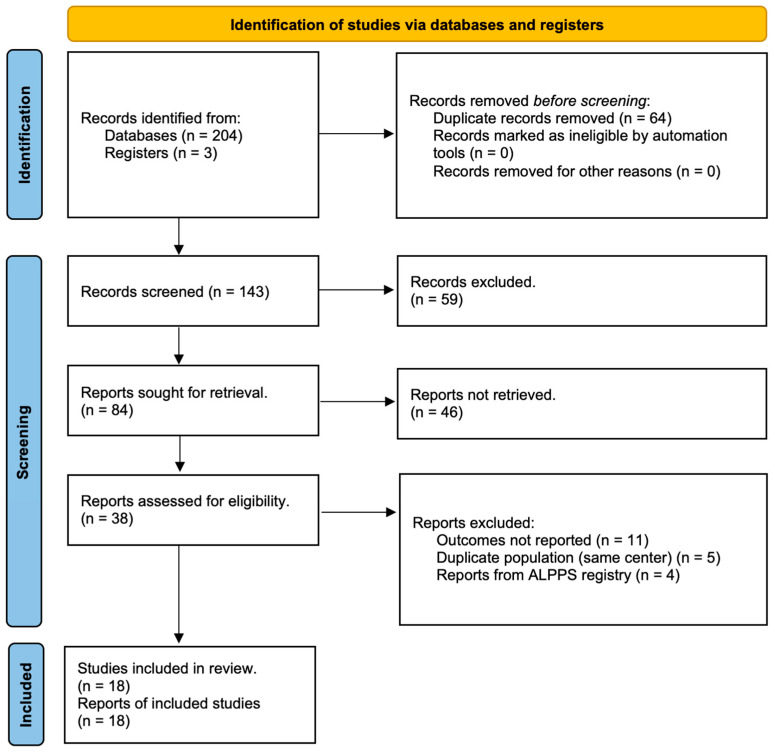
PRISMA flow chart of study selection.

**Figure 2 cancers-16-00771-f002:**
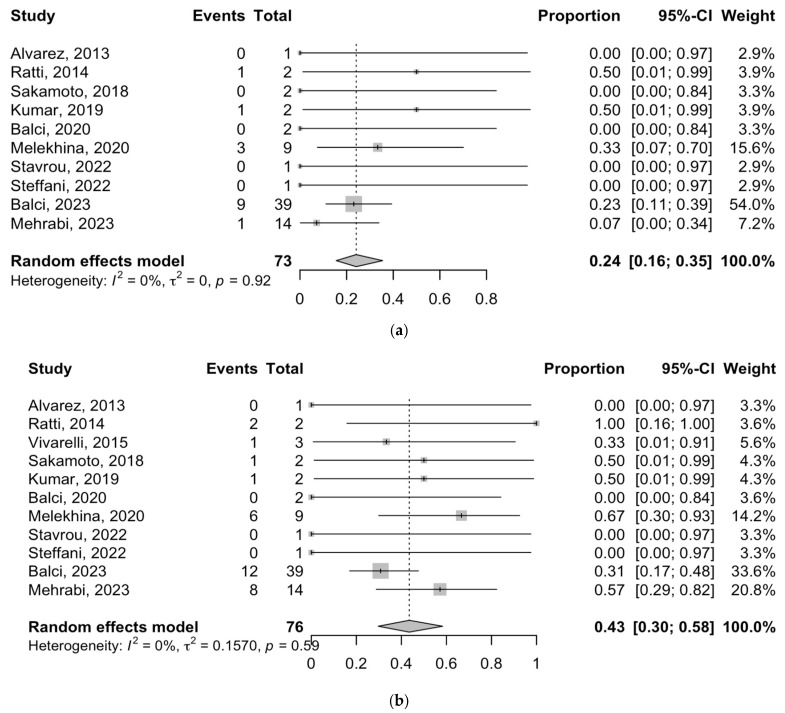
Forest plot of the major morbidities after first (**a**) and second (**b**) stages of ALPPS [10,21,23,25,28,29,30,31,34,35].

**Figure 3 cancers-16-00771-f003:**
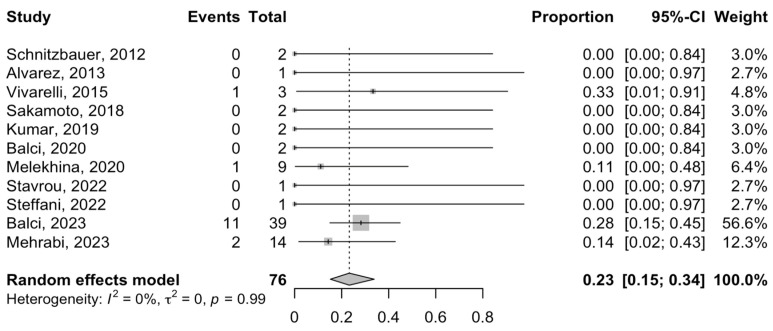
Forest plot of the PHLF after ALPPS procedure in patients with phCC [7,10,20,21,25,28,29,30,31,34,35].

**Figure 4 cancers-16-00771-f004:**
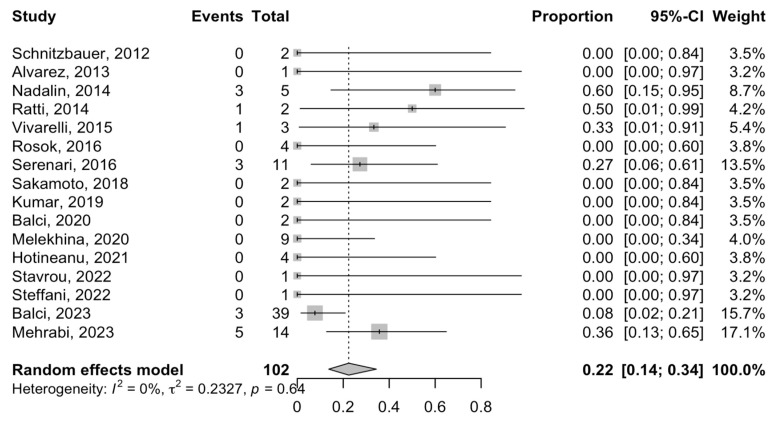
Forest plot of the mortality rates after ALPPS procedure in patients with phCC [7,10,20,21,22,23,25,26,27,28,29,30,31,33,34,35].

**Figure 5 cancers-16-00771-f005:**
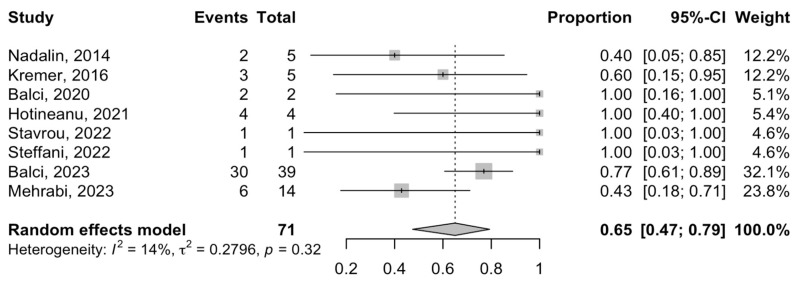
Forest plot of one-year disease-free survival rates after ALPPS procedure in patients with phCC [20,21,22,24,30,33,34,35].

**Figure 6 cancers-16-00771-f006:**
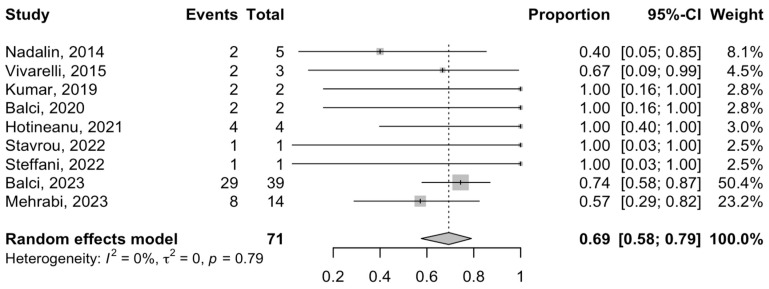
Forest plot of one-year overall survival rates after ALPPS procedure in patients with phCC [20,21,22,24,25,30,33,34,35].

**Table 1 cancers-16-00771-t001:** Baseline characteristics of the included studies and preoperative data of the patients.

Author, Year	Country	Study Period	Sample Size	Mean Age	Indication						
					CRLM	HCC	ihCC	phCC	GBC	Metastasis	Other
Schnitzbauer 2012 [7]	Germany	2007–2011	25	63	-	3	2	2	1	16	1
Alvarez 2013 [10]	Argentina	2011–2012	15	54	10	1	-	1	-	3	-
Nadalin 2014 [22]	Germany	2010–2013	15	67	5	1	4	5	-	-	-
Ratti 2014 [23]	Italy	2012	8	58	5	-	-	2	1	-	-
Vivarelli 2015 [25]	Italy	2013–2014	9	60	4	1	1	3	-	-	-
Kremer 2016 [24]	Germany	2011–2014	19	57	11	-	2	5	1	-	-
Rosok 2016 [26]	Norway, Sweden	2012–2015	36	67	25	4	-	4	-	-	3
Serenari 2016 [27]	Italy	2012–2014	50	62	-	8	8	11	1	22	-
Sakamoto 2018 [28]	Japan	2015–2017	3	67	-	-	-	2	-	-	1
Kumar 2019 [29]	Singapore	2014–2019	8	61	6	-	-	2	-	-	-
Balci 2020 [30]	Turkey	2012–2019	2	53	-	-	-	2	-	-	-
Melekhina 2020 [31]	Russia	2013–2018	11	58	-	-	-	11	-	-	-
Chebaro 2021 [32]	France	2011–2020	85	62	73	1	3	3	2	-	3
Hotineanu 2021 [33]	Moldova	2018–2020	18	62	7	6	-	4	-	1	-
Stavrou 2022 [34]	Germany	2018	2	67	-	-	-	1	-	-	1
Steffani 2022 [35]	Germany	2019	4	NA	-	1	-	1	-	-	2
Balci 2023 [21]	Turkey (multicentric)	2010–2020	39	60.5	-	-	-	39	-	-	-
Mehrabi 2023 [20]	Germany	2011–2021	21 (30 ^^^)	64.1	-	-	7	14	-	-	-

HCC: hepatocellular carcinoma, ihCC: intrahepatic cholangiocarcinoma, phCC: perihilar cholangiocarcinoma, GBC: gall bladder cancer, CRLM: colorectal liver metastasis, NA: not available. ^^^ Nine patients with phCC were excluded in order to avoid data duplication.

**Table 2 cancers-16-00771-t002:** Assessment of the quality of included studies.

Authors	Q1	Q2	Q3	Q4	Q5	Q6	Q7	Q8	Score (Quality)
Schnitzbauer 2012 [7]	2	0	0	2	0	2	2	0	8 (intermediate)
Alvarez 2013 [10]	2	0	0	2	0	2	2	0	8 (intermediate)
Nadalin 2014 [22]	2	0	0	2	0	2	2	0	8 (intermediate)
Ratti 2014 [23]	2	0	0	2	0	2	1	0	7 (low)
Vivarelli 2015 [25]	2	0	0	2	0	2	2	0	8 (intermediate)
Kremer 2016 [24]	2	2	0	2	1	2	2	0	11 (intermediate)
Rosok 2016 [26]	2	1	0	2	0	2	2	0	9 (intermediate)
Serenari 2016 [27]	2	0	0	2	0	2	1	0	7 (low)
Sakamoto 2018 [28]	2	0	0	2	0	1	0	0	5 (low)
Kumar 2019 [29]	2	0	2	2	0	1	0	0	7 (low)
Balci 2020 [30]	2	0	0	2	0	2	1	0	7 (low)
Melekhina 2020 [31]	2	2	0	2	0	2	0	0	8 (intermediate)
Chebaro 2021 [32]	2	2	0	2	0	2	0	0	8 (intermediate)
Hotineanu 2021 [33]	2	0	2	2	0	2	1	0	9 (intermediate)
Stavrou 2022 [34]	2	0	0	2	0	2	1	0	7 (low)
Steffani 2022 [35]	2	0	0	2	0	2	2	0	8 (intermediate)
Balci 2023 [21]	2	2	0	2	0	2	2	0	10 (intermediate)
Mehrabi 2023 [20]	2	2	0	2	0	2	2	0	10 (intermediate)

0 = not reported; 1 = reported but inadequate; 2 = reported and adequate. >12 = high; 8–12 = intermediate; <8 = low. Q1: Did the study have a clearly stated aim? Q2: Were consecutive patients included? Q3: Were data collected prospectively? Q4: Were endpoints appropriate to the study? Q5: Was there an unbiased assessment of endpoints? Q6: Was the follow-up period adequate? Q7: Was there loss to follow-up <5%? Q8: Was there a prospective calculation of study size?

**Table 3 cancers-16-00771-t003:** Perioperative data of the patients with phCC that underwent ALPPS procedure.

Author, Year	phCC (%)	Type of ALPPS	Type of Resection	BDA (n)	BDA (Stage I or II)
Schnitzbauer 2012 [7]	2 (8)	Classic	Right trisectionectomy	Yes	2nd
Alvarez 2013 [10]	1 (6.7)	Classic	Right trisectionectomy	Yes (1)	1st
Nadalin 2014 [22]	5 (33.3)	Classic	Right trisectionectomy	Yes (5)	2nd
Ratti 2014 [23]	2 (25)	Classic	Right trisectionectomy	Yes (2)	1st
Vivarelli 2015 [25]	3 (33.3)	Classic	NA	Yes (3)	1st
Kremer 2016 [24]	5 (26.3)	Classic	Right trisectionectomy	Yes (5)	NA
Rosok 2016 [26]	4 (11.1)	Classic	NA	No	NA
Serenari 2016 [27]	11 (22)	Classic	Right trisectionectomy	No	NA
Sakamoto 2018 [28]	2 (66.7)	Partial TIPE ALPPS	1 right hepatectomy, 1 left trisectionectomy	Yes (2)	NA
Kumar 2019 [29]	2 (25)	Partial	Right trisectionectomy	No	NA
Balci 2020 [30]	2 (100)	Laparoscopic Partial	Right trisectionectomy	Yes (2)	2nd
Melekhina 2020 [31]	9 (100)	PRALPPS	NA	Yes (9)	2nd
Chebaro 2021 [32]	3 (3.5)	Classic	NA	NA	NA
Hotineanu 2021 [33]	4 (22)	Classic/Anterior/Partial	Right trisectionectomy	Yes (4)	1st
Stavrou 2022 [34]	1 (50)	Partial	Right trisectionectomy	Yes (1)	1st
Steffani 2022 [35]	1 (25)	Soft-ALPPS	NA	NA	NA
Balci 2023 [21]	39 (100)	Classic/Modified	12 right hepatectomy, 27 left trisectionectomy	Yes	1st (8 patients) 2nd (31 patients)
Mehrabi 2023 [20]	14 (67)	Classic/Modified	4 right hepatectomy, 10 left trisectionectomy	Yes	2nd

BDA: biliodigestive anastomosis, PRALPPS: percutaneous radiofrequency-assisted liver partition with portal vein embolization in staged liver resection, Soft-ALPPS: PVE + non-absorbable foil. NA: not available.

**Table 4 cancers-16-00771-t004:** Postoperative and oncological outcomes of the patients with phCC that underwent ALPPS procedure.

Author, Year	phCC (%)	R0 n (%)	Major Morbidity n (%)		PHLF n (%)	Mortality n (%)	1-yr DFS n (%)	1-yr OS n (%)
			Stage I	Stage II				
Schnitzbauer 2012 [7]	2 (8)	2 (100)	NA	NA	0 (0)	0 (0)	NA	NA
Alvarez 2013 [10]	1 (6.7)	1 (100)	0 (0)	0 (0)	0 (0) ^a^	0 (0)	NA	NA
Nadalin 2014 [22]	5 (33.3)	4 (80)	NA	NA	NA	3 (60)	2 (40)	2 (40)
Ratti 2014 [23]	2 (25)	2 (100)	1 (50)	2 (100)	NA	1 (50)	NA	NA
Vivarelli 2015 [25]	3 (33.3)	3 (100)	NA	1 (33)	1 (33) ^b^	1 (33)	NA	2 (67)
Kremer 2016 [24]	5 (26.3)	5 (100)	NA	NA	NA	NA	3 (60)	NA
Rosok 2016 [26]	4 (11.1)	3 (75)	NA	NA	NA	0 (0)	NA	NA
Serenari 2016 [27]	11 (22)	11 (100)	NA	NA	NA	3 (27)	NA	NA
Sakamoto 2018 [28]	2 (66.7)	NA	0 (0)	1 (50)	0 (0)	0 (0)	NA	NA
Kumar 2019 [29]	2 (25)	NA	1 (50)	1 (50)	0 (0)	0 (0)	NA	2 (100)
Balci 2020 [30]	2 (100)	NA	0 (0)	0 (0)	0 (0)	0 (0)	2 (100)	2 (100)
Melekhina 2020 [31]	9 (100)	8 (89)	3 (27)	6 (67)	1 (11) ^a^	0 (0)	NA	NA
Chebaro 2021 [32]	3 (3.5)	NA	NA	NA	NA	NA	NA	NA
Hotineanu 2021 [33]	4 (22)	4 (100)	NA	NA	NA	0 (0)	4 (100)	4 (100)
Stavrou 2022 [34]	1 (50)	1 (100)	0 (0)	0 (0)	0 (0)	0 (0)	0 (0)	0 (0)
Steffani 2022 [35]	1 (25)	1 (100)	0 (0)	0 (0)	0 (0) ^a^	0 (0)	0 (0)	0 (0)
Balci 2023 [21]	39 (100)	32 (82)	9 (23)	12 (31)	11 (28) ^a^	3 (7.6)	30 (36)	29 (36)
Mehrabi 2023 [20]	14 (67)	10 (71)	1 (7.1)	8 (57)	2 (14.3) ^a^	5 (36)	6 (43)	8 (57)

phCC: perihilar cholangiocarcinoma, PHLF: post-hepatectomy liver failure, OS: overall survival, DFS: disease-free survival, ^a^ based on the ISGLS classification, ^b^ based on the 50–50 criteria, NA: not available.

## Data Availability

The data are available upon a reasonable request.

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
