# Peer review of "Promising Results of Associating Liver Partition and Portal Vein Ligation for Staged Hepatectomy for Perihilar Cholangiocarcinoma in a Systematic Review and Single-Arm Meta-Analysis"

_cancers, 2024, doi:10.3390/cancers16040771_

Round 1

Reviewer 1 Report

Comments and Suggestions for Authors

This study conducted a meta-analysis of ALPPS for perihilar cholangiocarcinoma, and reviewed the morbidity, mortality, PHLF, one-year overall-and disease-free survival. This study included 112 phCC patients from 18studies.

This study may be valuable and provide useful information to hepatobiliary surgeons. However, there are some limitations.

1.       The definition of PHLF is unclear. Table 4 shows the outcomes pf patients. One center (Vivarelli) adopted 50-50 criteria. Other centers used ISGLS to define PHLF? The grading (A,B,C) of PHLF is missing.

2.       The rate of PHLF was 9% in this review. The rate is quite low. However, the mortality rate was 14%. What was the cause of mortality? Liver failure? Sepsis? The authors should refer to the cause of mortality, if data regarding the causes of mortality are available.

3.       The future liver volume rate was also missing. If possible, the authors should describe future liver volume. FLV with less than 30% may be an indication for ALPPS. How many cases with FLV of less than 30% were performed in this review of 112 cases? Was there any association of small FLV with PHLF and outcomes?

Author Response

Reviewer 1:

This study conducted a meta-analysis of ALPPS for perihilar cholangiocarcinoma, and reviewed the morbidity, mortality, PHLF, one-year overall-and disease-free survival. This study included 112 phCC patients from 18studies. This study may be valuable and provide useful information to hepatobiliary surgeons. However, there are some limitations.

  1. The definition of PHLF is unclear. Table 4 shows the outcomes pf patients. One center (Vivarelli) adopted 50-50 criteria. Other centers used ISGLS to define PHLF? The grading (A,B,C) of PHLF is missing.

Thank you for your constructive comment. The definition of PHLF is now provided in the Supplementary Table and cited accordingly in the manuscript (Line 215).

  1. The rate of PHLF was 9% in this review. The rate is quite low. However, the mortality rate was 14%. What was the cause of mortality? Liver failure? Sepsis? The authors should refer to the cause of mortality, if data regarding the causes of mortality are available.

Thank you very much for your consideration. We have reassessed the all manuscripts in means of the rate of PHLF and reanalysed the data in this regard. carried out the analyses. Cause of mortality is also reported separately for each study in the Supplementary Table, which is also cited in the manuscript.

  1. The future liver volume rate was also missing. If possible, the authors should describe future liver volume. FLV with less than 30% may be an indication for ALPPS. How many cases with FLV of less than 30% were performed in this review of 112 cases? Was there any association of small FLV with PHLF and outcomes?

Thank you very much for your accurate review. We have now added the inclusion criteria in each study in the Supplementary table, concerning the phCC patients undergone ALPPS. The table has been cited in the manuscript, respectively.

Reviewer 2 Report

Comments and Suggestions for Authors

This is a systematic review and meta-analysis of results of patients with perihilar cholangiocarcinoma after Associating Liver Partition and Portal vein Ligation for Staged hepatectomy. It reviews the appropriate literature and is well thought out and organized.

I only have a few comments.

1. What do the authors suggest for patients with perihilar cholangiocarcinomas close to the biliary confluence. Do they have any insights on whether or not there are any scenarios were ALPPS could help this patient population? Or is there no role for ALPPS in all of these patients?

2. Laparoscopic resection of perihilar cholanciocarcinoma has been reported. Can the authors comment on this approach?

Gumbs AA, Jarufe N, Gayet B. Minimally invasive approaches to extrapancreatic cholangiocarcinoma. Surg Endosc. 2013 Feb;27(2):406-14. doi: 10.1007/s00464-012-2489-8. Epub 2012 Aug 28. PMID: 22926892.

3. Similarly, robotic resections of perihilar cholangiocarcinoma has been described. Can the authors comment on this?

Cillo U, D'Amico FE, Furlanetto A, Perin L, Gringeri E. Robotic hepatectomy and biliary reconstruction for perihilar cholangiocarcinoma: a pioneer western case series. Updates Surg. 2021 Jun;73(3):999-1006. doi: 10.1007/s13304-021-01041-3. Epub 2021 Apr 16. PMID: 33861401; PMCID: PMC8184707.

4. Artificial intelligence with augmented reality and enhanced navigation may have some advantages when compared to the standard robotic approach. Can the authors comment? What do they think of biophotonics?

Gumbs AA, Gayet B. Why Artificial Intelligence Surgery (AIS) is better than current Robotic-Assisted Surgery (RAS). Art Int Surg 2022;2:207-12. http://dx.doi.org/10.20517/ais.2022.41

Author Response

  1. What do the authors suggest for patients with perihilar cholangiocarcinomas close to the biliary confluence. Do they have any insights on whether or not there are any scenarios were ALPPS could help this patient population? Or is there no role for ALPPS in all of these patients?

Your precise point of view is greatly appreciated by the authors. Our response regarding this valuable comment is now provided in the discussion, as follows:

“Technically, ALPPS is also a feasible method in patients with phCC near the biliary confluence, since hepaticojejunostomy after confluence resection was carried out almost in 70% of the cases. Particularly, among patients with phCC suffering from preoperative cholestase and cholangitis, a delayed bile duct reconstruction during the 2nd stage of ALPPS is shown to diminish the rate of morbidity and mortality.” (Page 19, Line 74-80)

  1. Laparoscopic resection of perihilar cholanciocarcinoma has been reported. Can the authors comment on this approach?

Gumbs AA, Jarufe N, Gayet B. Minimally invasive approaches to extrapancreatic cholangiocarcinoma. Surg Endosc. 2013 Feb;27(2):406-14. doi: 10.1007/s00464-012-2489-8. Epub 2012 Aug 28. PMID: 22926892.

We appreciate your knowledgeful point of view. The suggested articles have now been cited and discussed in the discussion, as follows:

“In addition, recent studies have highlighted the advantageous role of minimally invasive surgery in treating patients with phCC. A recent series of laparoscopic and robot-assisted liver resections among these patients revealed no major complications and mortality [57,58]. Based on this evi-dence, minimally invasive techniques can be considered safe and feasible approaches, at least for the first stage of ALPPS, to reduce the extent of the procedure and minimize potential complications.” (Page 19-20, Line 119-20)

  1. Similarly, robotic resections of perihilar cholangiocarcinoma has been described. Can the authors comment on this?

Cillo U, D'Amico FE, Furlanetto A, Perin L, Gringeri E. Robotic hepatectomy and biliary reconstruction for perihilar cholangiocarcinoma: a pioneer western case series. Updates Surg. 2021 Jun;73(3):999-1006. doi: 10.1007/s13304-021-01041-3. Epub 2021 Apr 16. PMID: 33861401; PMCID: PMC8184707.

Thank you very much for your comment. The suggested articles have now been cited and discussed in the discussion, as follows:

“In addition, recent studies have highlighted the advantageous role of minimally invasive surgery in treating patients with phCC. A recent series of laparoscopic and robot-assisted liver resections among these patients revealed no major complications and mortality [57,58]. Based on this evidence, minimally invasive techniques can be considered safe and feasible approaches, at least for the first stage of ALPPS, to reduce the extent of the procedure and minimize potential complications. Furthermore, numerous advanced modalities, such as augmented reality and enhanced tumor navigation via biophotonics (such as indocyanine green), have been integrated into liver surgery, yielding improved results. Particularly within the realm of robotic surgery, these options can enhance their utilization and amplify their benefits. Potential advantages encompass reduced morbidity and enhancements in oncological outcomes via improved tumor navigation. However, recent surveys have revealed certain limitations, including inadequate technological development and a lack of clinical evidence.” (Page 19-20, Line 113-128).

  1. Artificial intelligence with augmented reality and enhanced navigation may have some advantages when compared to the standard robotic approach. Can the authors comment? What do they think of biophotonics?

Gumbs AA, Gayet B. Why Artificial Intelligence Surgery (AIS) is better than current Robotic-Assisted Surgery (RAS). Art Int Surg 2022;2:207-12. http://dx.doi.org/10.20517/ais.2022.41

Despite the hopes for surgeons on issues related to AI, surgical teams should receive additional training in AI. Given that this inclusion of the basics of AI in their curricula has not yet happened, we should wait for more data to discuss the results. The suggested articles have now been cited and discussed in the discussion, as follows:

“Based on this evidence, minimally invasive techniques can be considered safe and feasible approaches, at least for the first stage of ALPPS, to reduce the extent of the procedure and minimize potential complications. Furthermore, numerous advanced modalities, such as augmented reality and enhanced tumor navigation via biophotonics (such as indocyanine green), have been integrated into liver surgery, yielding improved results. Particularly within the realm of robotic surgery, these options can enhance their utilization and amplify their benefits. Potential advantages encompass reduced morbidity and enhancements in oncological outcomes via improved tumor navigation. However, recent surveys have revealed certain limitations, including inadequate technological development and a lack of clinical evidence.” (Page 19-20, Line 116-128).

Reviewer 3 Report

Comments and Suggestions for Authors

The manuscript is well organized and touching all the important parts of the learning curve in ALPPS such as TNM, icterus, infection and regenarative potential of the liver and thus reflecting the learning curve since the first report by Schnitzbauer in 2012.

Author Response

The manuscript is well organized and touching all the important parts of the learning curve in ALPPS such as TNM, icterus, infection and regenarative potential of the liver and thus reflecting the learning curve since the first report by Schnitzbauer in 2012.

Thank you for your interest in our manuscript.

Round 2

Reviewer 1 Report

Comments and Suggestions for Authors

I think that this manuscript is acceptable in this form.